# Toughening of Epoxy Resin: The Effect of Water Jet Milling on Worn Tire Rubber Particles

**DOI:** 10.3390/polym11030529

**Published:** 2019-03-20

**Authors:** Peter Tamas-Benyei, Eniko Bitay, Hajime Kishi, Satoshi Matsuda, Tibor Czigany

**Affiliations:** 1Department of Polymer Engineering, Faculty of Mechanical Engineering, Budapest University of Technology and Economics, Muegyetem rkp. 3., H-1111 Budapest, Hungary; czigany@eik.bme.hu; 2MTA-BME Research Group for Composite Science and Technology, Muegyetem rkp. 3., H-1111 Budapest, Hungary; 3Faculty of Technical and Human Sciences, Sapientia Hungarian University of Transylvania, 540485 Târgu-Mureş, Op. 9., Cp. 4., Romania; bitay.eniko@eme.ro; 4Department of Chemical Engineering, Graduate School of Engineering, University of Hyogo, 8 Chome-2-1 Gakuen Nishimachi, Nishi Ward, Kobe 651-2103, Japan; kishi@eng.u-hyogo.ac.jp (H.K.); smatsuda@eng.u-hyogo.ac.jp (S.M.)

**Keywords:** epoxy, toughening, ground tire rubber, rubber particle

## Abstract

In this work a cycloaliphatic amine-cured epoxy (EP) resin was modified by micron-scale rubber particles (RP). Nominal RP, in sizes of 200 and 600 µm respectively, were produced using worn truck tires and ultra-high-pressure water jet cutting. The RP were dispersed into the EP resin using different mixing techniques (mechanical, magnetic, and ultrasonic stirring) prior to the introduction of the amine hardener. The dispersion of the RP was studied using optical light microscopy. A longer mixing time reduced the mean size of the particles in the EP compounds. Static (tensile and flexural), dynamic (unnotched Charpy impact), and fracture mechanical (fracture toughness and strain-energy release rate) properties were determined. The incorporation of the RP decreased the stiffness and strength values of the modified EPs. In contrast, the irregular and rough surface of the RP resulted in improved toughness. The fracture toughness and strain-energy release rate were enhanced up to 18% owing to the incorporation of 1% by weight (wt%) RP. This was traced to the effects of crack pinning and crack deflection. Considerably higher improvement (i.e., up to 130%) was found for the unnotched Charpy impact energy. This was attributed to multiple cracking associated with RP-bridging prior to final fracture.

## 1. Introduction

In the last decades, polymers and their composites received great attention due to their advantageous and tailorable properties [1]. For high-performance thermoset composites the most widely applied resin is epoxy (EP). Mechanical properties of EP resins are usually much better than that of the commonly used resins. In order to improve their inherent brittleness, a characteristic feature for all thermosets, different approaches are followed. The related approaches can be divided into two groups, chemical modification by reactive additives or physical toughening by fillers. Chemical additives that create molecular connections and perfect dispersion cause homogenous properties. A disadvantage of the method using chemical additives is that the low crosslinking rate causes lower mechanical properties. The physical toughening method can be divided into three groups on the basis of the filler material: inorganic particles, liquid rubbers, and thermoplastic fillers. The most often applied of these materials is functionalized liquid rubber. First in 1968 McGarry and Willner [2] and then in 1973 Sultan and McGarry [3] use functionalized liquid rubber to improve toughness of resins. Usually, liquid rubbers are functionalized [4,5] by interacting chemical reactions in EP resins and taking shape by phase-separation [6,7]. The literature overview showed that this field is commonly examined [8,9,10,11,12,13,14]. Hwang et al. [15] investigated size distribution of rubber and its effect on tensile properties of EP resin. Their results showed that the average size of rubber particles (RP) was 2.4 µm. The addition of rubber decreased tensile strength and Young’s modulus, but increased fracture toughness and strain-energy release rate significantly. The addition of rubber can toughen EP polymers by particle cavitation/matrix shear banding and plastic void growth. Pearson et al. [16] investigated the influence of particle size and particle size distribution of rubber modified EP. On the basis of their results the optimal size of RP in the case of liquid rubber was 0.1 µm, and in the case of solid rubber it was less than 100 µm. Ricciardi et al. [17] used nitrile rubber to toughen glass fiber reinforced EP composites. According to their results the modified composites showed smaller delamination, although the absorbed energy was the same and the load was higher. The delaminated area of the modified samples was less than the neat one although the absorbed energy result was the same in each case. Karger-Kocsis and Friedrich [18,19] investigated fatigue crack propagation of carboxyl-terminated acrylonitrile-butadiene rubber (CTBN) and silicon rubber (SI) modified EP resin. The incorporation of CTBN and/or SI dispersion in the EP matrix improved the resistance to fatigue crack propagation. In these studies, the particle size of the rubbers was between 0.1 and 1 µm. The simultaneous use of both modifiers, i.e., CTBN and SI produced a multiplicative, synergistic effect and improved the fatigue resistance of the materials.

Quan et al. [20] explained and modelled the toughening mechanism of RP in their numerical and experimental studies. Fatigue behavior [21,22] and modelling [23,24,25] of liquid rubber modified EP has been examined by many researchers, which is important to aspects of engineering application. 

Difficulties during the applications of liquid rubbers have led researchers to incorporate vulcanized RP into EP. The milling of worn tires is one method to obtain vulcanized particles. However, the size of recycled ground tire rubber (GTR) is several times greater than the liquid rubbers mentioned above. Nonetheless, modification of EP using ground RP is of great practical relevance due to environmental aspects. Firstly, Rodriguez [26] investigated the influence of cryogenically ground rubber (CGR) on the mechanical properties of unsaturated polyester (UP) resin. The specimens were prepared using two particle size fractions (250 and 150 µm). The particles were treated by silane agents to improve adhesion between the particles and the UP. On the basis of their results, the glass transition temperature was increased by the addition of CGR, but stiffness and strength in both tension and flexure were decreased significantly. The silane coupling agents improved the tensile strength. Bagheri et al. [27] used recycled RP, as well as CTBN, for toughening of EP. They reported synergistic toughening when 2.5 phr of RP and 7.5 phr of CTBN were incorporated. The results illustrated that whereas the recycled RP improved fracture resistance of the EP resin by less than 20%, the CTBN rubber enhanced the fracture toughness by greater than 200%. The improvement was explained by plastic zone branching mechanism that enlarged the effective damage zone size and further enhanced the crack growth resistance. Boynton and Lee [28] analyzed the influence of recycled elastomer particles and CTBN on the fracture properties of EP in compact tension tests. The modified EP showed a decrease in fracture toughness and an increase in strain rate. As well, the hybrid-toughened EPs not only have better fracture properties but should exhibit higher mechanical performance. Saglam et al. [29,30] used recycled GTR particles to improve the toughness of EP. According to their results the tensile strength and Young’s modulus decreased because of the addition of rubber filler (average size of particles was 400 µm) but the fracture toughness was markedly increased owing to the aminosilane surface treatment of GTR. Kaynak et al. [31] used different silane coupling agents (SCA) to improve the interfacial adhesion between EP and recycled RP interface, and they investigated the mechanical properties of the corresponding EP materials. Tensile and impact properties, including fracture toughness, were prominently decreased by the addition of RP. The use of SCAs for surface treatment of RP did not effectively improve the impact and fracture toughness of the EP specimens. The smallest reduction in fracture toughness was obtained by three aminopropyl triethoxysilane treatments of RPs. Celikbilek et al. [32] modified EP using solid and liquid RP and found that the impact and fracture toughness decreased by the addition of RP. When EP was modified using solid RP and liquid elastomer, no significant synergistic effect was observed in the mechanical performance of the samp1es. Abadyan et al. [33] studied the fracture behavior of a hybrid-rubber modified EP system. The modified EP contained amine-terminated butadiene acrylonitrile (ATBN) rubber and recycled tire particles. Testing of the blends revealed synergistic toughening when 2.5 phr GTR and 7.5 phr ATBN were incorporated. Karger-Kocsis et al. [34] reviewed and summarized the application of GTR particles in thermoplastics, thermosets, and rubbers. According to their paper GTR was not applicable for efficient toughening of EP resins. The main reason for this claim was the smooth-cut surface of the particles. Aliabdo et al. [35] investigated the utilization of waste rubber in non-structural applications. Three types of RP were used as filler, one crumb rubber (4 mm) and two fibrous particles of different lengths (2.36 and 1.18 mm). The authors found that the compressive strength of the investigated materials increased by adding finer fibrous RP filler as compared with crumb particle filler. Flexural strength was decreased by adding rubber filler. A positive influence of finer particles was proven by cyclic compression tests. In the case of finer fillers, the number of cycles until failure was three times greater than that of coarser particles. Irez et al. [36] analyzed the influence of recycled and devulcanized RP on the mechanical properties of EP-based composites. Their results revealed that both the adhesion between the particles and EP resin and the homogenous dispersion had key roles in the toughening mechanism. As well, the relevance of rubbers and the recycling of rubbers has been shown by different researchers [37,38].

The data in the literature listed above seems to confirm that RP are not applicable effectively for EP toughening due to the smooth surface of the particles. Today, a new tire recycling method is gaining importance: the water jet milling (WJM) method. By applying the WJM method, RP were made using ground truck tires. Instead of conventional shredding and mechanical grinding of tires, this technology only applies an ultra-high-pressure water jet for extraction and simultaneous milling of rubber parts, thereby leaving the reinforcements intact. The range of sizes of RP produced by this method were between 50 and 500 µm and between 500 and 1500 µm. The nominal size of RP used in this work were between 200 and 600 µm. The WJM method causes the surface of the particles to be irregular and possess a relatively high specific surface area (Figure 1). Our working hypothesis is that better improvement in the toughness of RP can be reached by the incorporation of WJM than that by conventional ground versions.

The aim of this paper is to report on how the incorporation of water jet milled RB influences the mechanical properties, and especially the toughness of EP resin.

## 2. Materials and Methods

The matrix material used was MR 3009 cycloaliphatic EP resin (IPOX Chemicals Kft., Hungary) and MH3120 modified cycloaliphatic amine hardener (IPOX Chemicals Kft., Hungary). The mixing ratio was stoichiometric, i.e., 100:20 by weight (wt%) MR 3009 to MH3120.

The RP were produced by Aquajet Ltd. (Budapest, Hungary) using the tread of used truck tires. The powder was not mixed side-wall and inner-liner and it was completely metal and textile free. The technical average particle size was about 400 µm. 

The optical microscopy (OM) study was executed using an Olympus BX-51 optical microscope (Hamburg, Germany) and Stream Software for picture analysis. A scanning electron microscopy (SEM) study was performed using a JEOL JSM 6380LA scanning electron microscope (Tokyo, Japan) and built-in picture analysis software. The size distribution of RP was calculated by measuring 100 particles per measurement on the fracture surface of broken Charpy specimens. Tensile tests were carried out according to ISO 527-4 standard by a Zwick Z020 universal tensile testing machine (Ulm, Germany) at a crosshead speed of 5 mm/min at room temperature. Tensile strength and Young’s modulus were calculated for five dumbbell samples 80 mm × 10 mm × 4 mm (length × width × thickness) besides the standard deviation. A flexural test was arranged according to ISO 14125 standard by a Zwick Z020 universal tensile testing machine (Ulm, Germany) at a crosshead speed of 5 mm/min at room temperature. The support distance was 64 mm. Flexural strength and flexural modulus were calculated as an average using five specimens 80 mm × 10 mm × 4 mm (length × width × thickness). The Charpy test was performed according to ISO 179 standard by a CEAST Resil Impact Junior pendulum (Torino, Italy) at 2 J energy and an angle of 150°. The support distance was 62 mm and the velocity of pendulum was 3.3 m/s. The Charpy impact strength was calculated for five unnotched specimens (80 mm × 10 mm × 4 mm (length × width × thickness). Compact tensile (CT) tests were done using the adaptation of ASTM D5045-99 standard and a Zwick Z005 universal tensile testing machine (Ulm, Germany) at a crosshead speed of 0.5 mm/min at room temperature. The plane-strain fracture toughness (K_IC_) (Equation (1)) and the critical strain-energy release rate (G_IC_) (Equation (2)) were calculated using the results of five tests on the specimens (Figure 2) and the dimension 60 mm × 60 mm × 4 mm (length × width × thickness). The Poisson’s ratio was measured during the CT tests using a Sobriety Mercury Monet DIC optical strain measurement system (Kourim, Czech Republic). The same equipment was used for the calculation of Young’s modulus. The notch of CT specimens was machined in the center of the sample using a Mutronic Diadisc 4200 diamond disc cutter (Rieden, Germany) and the sharp crack edge was prepared using a razor blade.
(1)$$K_{\mathrm{IC}}=\frac{F_{\max}}{BW^{1/2}}f\left(x\right)$$
(2)$$G_{\mathrm{IC}}=\frac{K_{\mathrm{IC}}}{E\left(1-\nu^{2}\right)\;}$$
where
(3)$$f\left(x\right)=\frac{\left(2+x\right)\left(0.886+4.64x-13.32x^{2}+14.72x^{3}-5.6x^{4}\right)}{{\left(1-x\right)}^{3/2}}\;$$
and 0.2 < *x* < 0.8:(4)x=a/W
where *K*_IC_ is the plane-strain fracture toughness (MPa·m^1⁄2^), *F_max_* is the highest load (N), *B* is the specimen thickness (mm), *W* is the specimen width (mm), *G_IC_* is the critical strain-energy release rate (kJ/m^2^), *E* is the Young’s modulus (MPa), *v* is the Poisson’s ratio (-), and *a* is the crack length (mm). 

## 3. Results

Experimental results of morphological and mechanical tests are discussed in this session.

### 3.1. Analysis of RP Size Distribution

Particle size distribution was measured using a CISA BA 200 electromagnetic sieve (Barcelona, Spain) according to the ISO 9276 standard. The results of the size dispersion measurements of RP are shown in Figure 3.

### 3.2. Analysis of Influence of Mixing Parameters on Particle Size

Firstly, three different mixing methods (magnetic, mechanical, and ultrasonic) were applied for different times (1, 2, 5, 15, 30, 60, 120 min) to analyze the influence of the type and the time of mixing on the particle size. The IKA MSH Basic mixer (Staufen, Germany) was used as a magnetic stirrer, the IKA RW 16 Basic mixer (Staufen, Germany) was used as a mechanical stirrer and the Bandelin Sonopuls HD 2070 mixer (Berlin, Germany) was used as an ultrasonic stirrer. For all methods of stirring we used the same type of DURAN^®^ glass container with a volume of 600 mL (ø70 mm and 100 mm height). After mixing, the curing agent was added into the EP/RP dispersion (1% wt% filler content) and mixed properly. The dispersion was poured into a polytetrafluoroethylene (PTFE) mold and was cured for 24 h at room temperature and later was treated for 2 h at 80 °C as post-curing. The cured samples were allowed to cool down slowly in the oven to room temperature. The RP dispersion was analyzed by OM study on cross-sections of damaged Charpy specimens. In every case at least 100 particles were measured. The results of the OM are shown in Figure 4. The results indicated that as the mixing time increased the mean size of the rubber agglomerates decreased. The explanation for this phenomenon is that the RP in the form of agglomerates disintegrated into smaller ones under high shear stresses.

This explanation was proven by comparing the effects of mechanical (MM), electromagnetic (EMM), and ultrasonic mixing (USM) on the particle size development. Note that MM generated higher shear forces than that of EMM and USM (Figure 3). The particle dimensions and dispersion were measured using an Olympus BX-51 optical microscope (Hamburg, Germany) and the Stream Software for picture analysis. We calculated the size distribution of RP by measuring 100 particles per measurement on the fracture surface of broken Charpy specimens. In every case, the particle size was reported as the largest dimensions of the particles using the built-in size determination method of the software. Figure 5 shows the particle size distribution. 

In Figure 6 the bright phase is the EP resin and the dark one is the rubber phase. It shows that in the case of the EMM and USM dispersions the sizes of particles were inhomogeneous, as compared with MM dispersion where the pictures show higher homogeneity. The smallest particle sizes were generated by mechanical stirring for 120 min. In the case of USM, after 5 min of stirring the material was remarkably degradated because of the generated high heat (~140 °C). On the basis of the results of the average size measurements for the applied mixing methods, MM was chosen as the preparation for samples used in further tests.

The influence of filler content on size distribution was also investigated using electron microscopy (Figure 7). Before measurement, the surface of the samples was coated with gold to provide conductivity. On the basis of our results, the average particle size increased as the filler content increased.

### 3.3. Influence of Filler Content on Mechanical Properties

The main aim of this study was to increase the toughening of EP resin by using RP. The toughening of materials was first investigated in Charpy tests. Figure 8 shows the Charpy impact strength as a function of the RP amount. Note that the Charpy impact strength was increased significantly due to the rubber filler, 1% wt% rubber filler caused greater than 130% improvement. As the filler content increased, impact strength decreased because untreated RP behaved as weak points. On the basis of the results, 1% wt% filler content was chosen for further tests. Most likely, the reason for this prominent increase was the multiple cracking associated with the RP-bridging effect before final fracture occurred.

As disclosed in the introduction, the RP decreased the tensile strength and Young’s modulus of the EP. The results of the tensile and flexural tests, shown in Table 1, confirm this behavior. The tensile strength and stiffness were reduced by 3% and 5% owing to the incorporation of only 1% wt% RP. Flexural strength and flexural modulus decreased due to the addition of 1% wt% rubber filler. The addition of RP caused a 20% and 10% decrease in flexural strength and flexural stiffness, respectively. The reason for these decreases was the presence of higher amounts of RP having lower statistical mechanical properties.

Using the results of compact tensile (CT) tests, the plane-strain fracture toughness and the critical strain-energy release rate were calculated (Figure 9). The Poisson’s ratio used was 0.34. Both values increased as rubber was added. Rubber filling enhanced fracture toughness and the strain-energy release rate by 8% and 19%, respectively. This behavior can be attributed to crack pinning and deflection caused by the RP. Crack pinning implies the deceleration of crack growth due to the RP particles acting as obstacles. This phenomenon has been explained by Karger-Kocsis et al. [18]. To prove our theory, we took SEM micrographs of the 1% wt% filled sample. Figure 10 shows one picture of the typical micrographs. It proves that RPs (indicated by the blue arrows) are able to retard and even stop crack growth (indicated by the red arrows).

## 4. Conclusions

Particle homogeneity and size distribution of the WJM on RP in EP was observed by using OM and SEM. It was found that by increasing mixing time the mean size of the RP agglomerates decreased. Examination of the micrographs show that the smallest average particle size (90 µm) was generated by mechanical mixing (MM), as compared with electromagnetic (EMM) and ultrasonic mixing (USM) methods. The explanation for this phenomenon is that RP created agglomerates and due to the higher shearing of MM agglomerates dispersed into smaller particles. Mechanical properties were examined by tensile, flexural, Charpy impact, and compact tensile (CT) tests. The results of the tensile and flexural tests showed that tensile strengths and Young modulus were decreased by RP, as well as flexural strength and flexural modulus were decreased by RP. The decrease was caused by bigger RP/agglomerates that behaved as local stress collecting zones. On the basis of the fracture test it can be stated that the Charpy impact strength was significantly increased by rubber modification (1% wt% filler content). The increase was greater than 130% as compared with the neat resin. The results of CT tests showed an 8% increase in plane-strain fracture toughness and a 19% increase in the critical strain-energy release rate. The reason for this behavior is that crack deflection may be considered by the RP, and moreover the RP retard the crack growing.

## Figures and Tables

**Figure 1 polymers-11-00529-f001:**
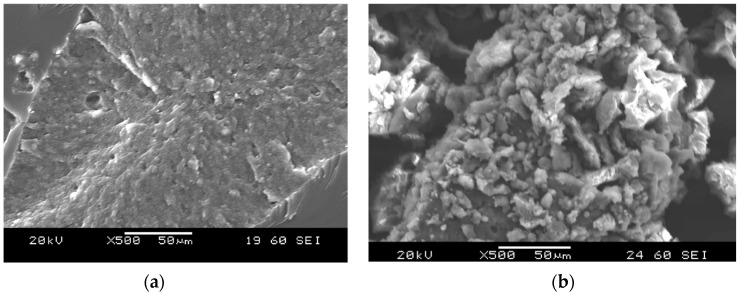
The surface of rubber particles (RP) prepared using a rotary mill (**a**) and using water jet milling (**b**).

**Figure 2 polymers-11-00529-f002:**
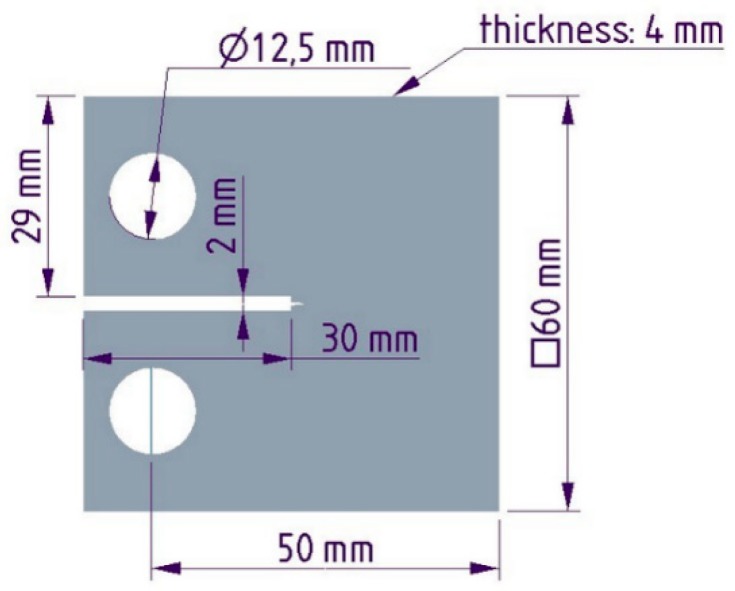
Dimensions of used CT samples.

**Figure 3 polymers-11-00529-f003:**
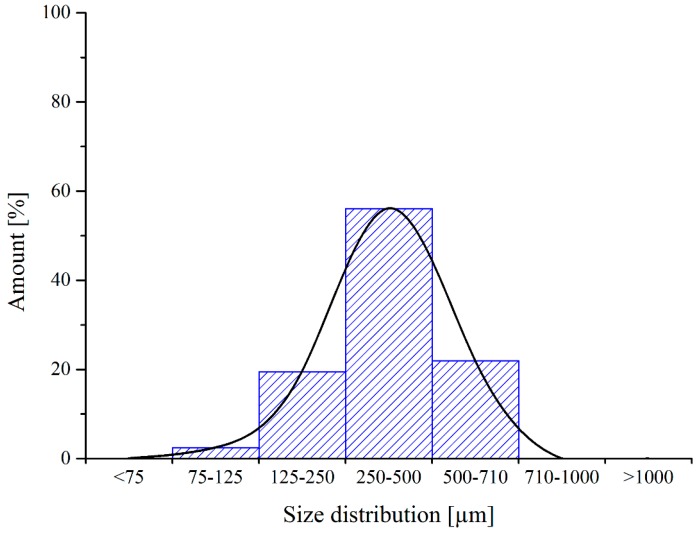
Size distribution of RP.

**Figure 4 polymers-11-00529-f004:**
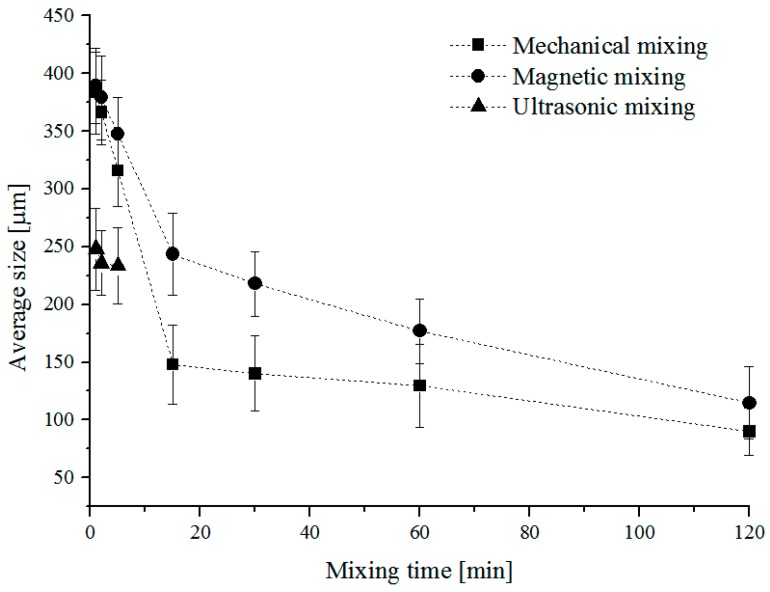
Average size of RP as a function of mixing time and methods.

**Figure 5 polymers-11-00529-f005:**
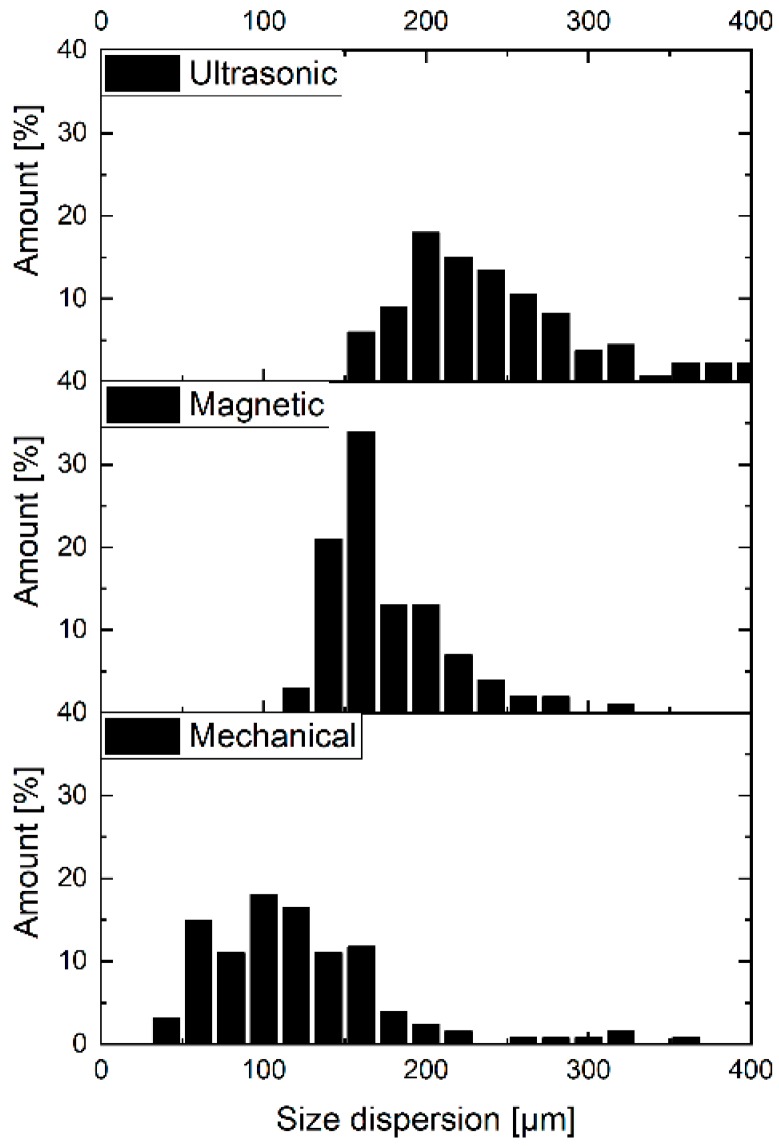
Particle size distribution at different stirring methods.

**Figure 6 polymers-11-00529-f006:**
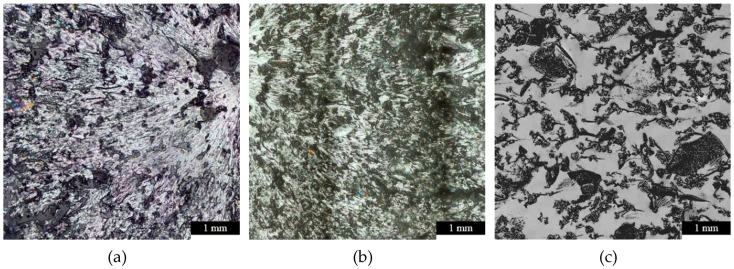
Different effects of mixing methods, (**a**) electromagnetic mixing (EMM) at lower shearing, (**b**) mechanical mixing (MM) at higher shearing, (**c**) ultrasonic mixing (USM).

**Figure 7 polymers-11-00529-f007:**
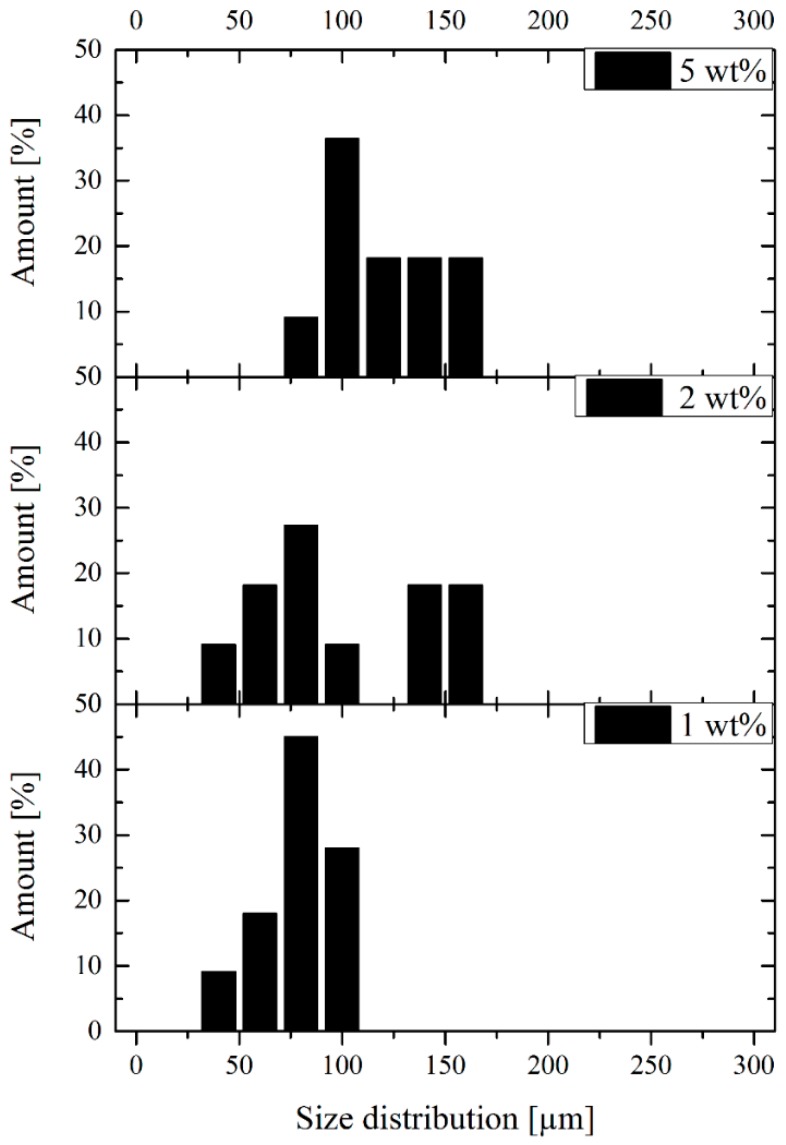
Influence of filler content on size distribution.

**Figure 8 polymers-11-00529-f008:**
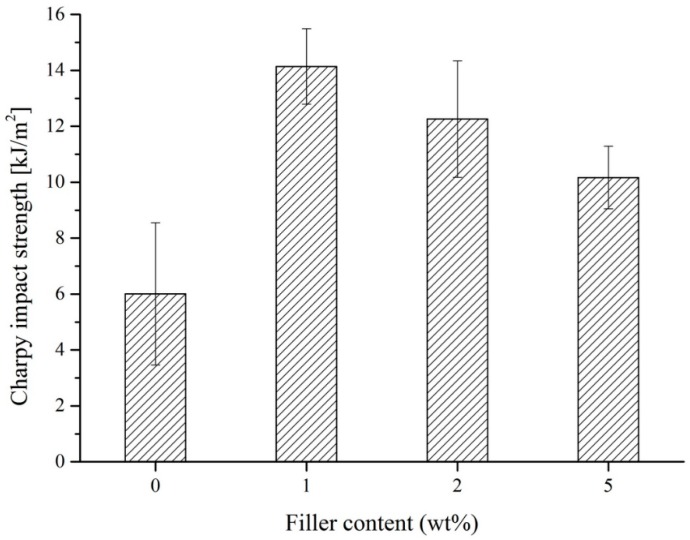
Charpy impact strength for rubber-filled epoxy (EP) and pure EP as a function of filler content.

**Figure 9 polymers-11-00529-f009:**
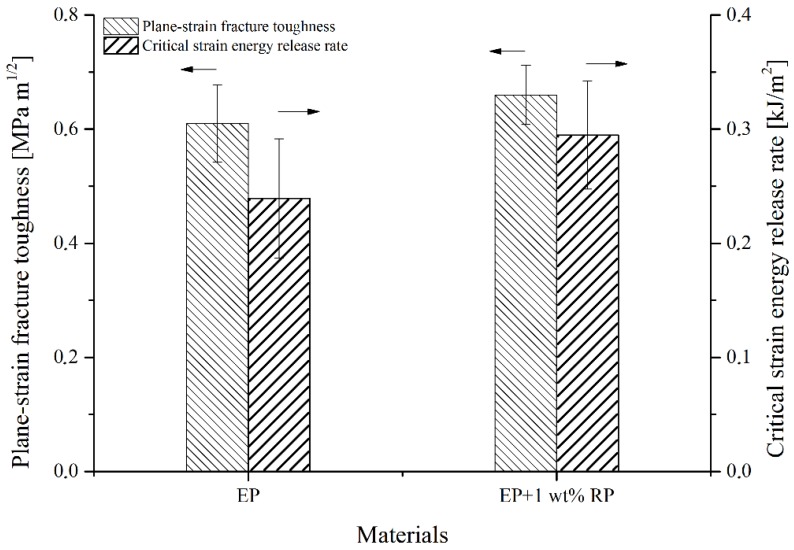
Results of CT tests of rubber-filled and pure epoxies.

**Figure 10 polymers-11-00529-f010:**
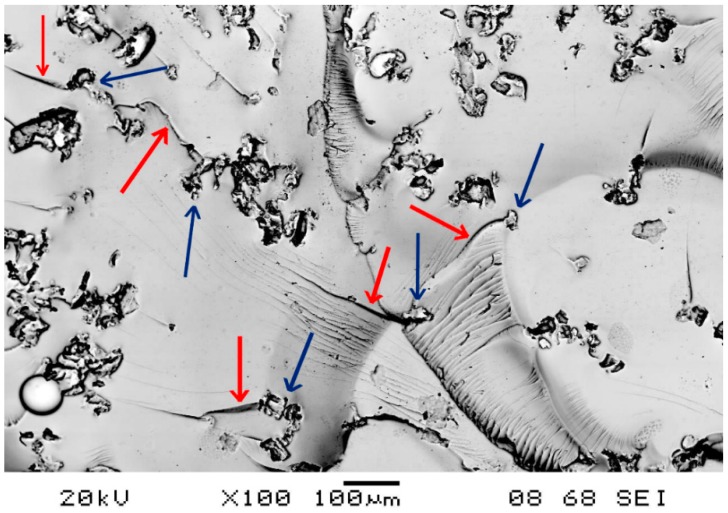
Crack propagation inside rubber-filled EP resin (RPs are indicated by the blue arrows, cracks are indicated by the red arrows).

**Table 1 polymers-11-00529-t001:** Results of tensile and three-point bending tests of rubber-filled EP and pure EP.

Materials	Tensile Strength [MPa]	Young’s Moduli [MPa]	Flexural Strength [MPa]	Flexural Moduli [MPa]
EP	38.11 ± 0.96	1551 ± 51	65.27 ± 1.40	1929 ± 124
EP + 1% wt% RP	36.84 ± 0.81	1472 ± 45	52.08 ± 5.17	1736 ± 29

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
