# Peer review of "Toughening of Epoxy Resin: The Effect of Water Jet Milling on Worn Tire Rubber Particles"

_polymers, 2019, doi:10.3390/polym11030529_

Round 1
Reviewer 1 Report
In the present paper,water jet milled worn tire rubber particles were used as additives for toughening of epoxy resin, and the effect for the properties of epxoy resins were characterized. It is good to find applications for waste tires.Some suggstions are as follows.
1.The size of the particle produced by water jet milled and the dispersity of partilce size are some of the key factors for toughening of epoxy resin. The size of particles is revealed in paper, then what is the dispersity of size of RP? Does the dispersity of size of RP affect the properties of EP?
2.3 mixing methods are adopted in the paper, and average size of rubber particles in function of mixing time and methods were shown in Figure 2.But for ultrasonic mixing, the data reported less than 5 minutes and for the temperature increased rapidly,what is the reason? If using solvents mixing, could it be working for longer time?
3.For USM method, the author piont that "In case of USM after 5 minutes stirring material was degradated 178 remarkably because of generated high heat (~140 °C)". What is the reaction? Any evidence for the degradation?
4.For Figure 2, what is the content of RP in epoxy resin?
5.For Figure 3, the title is Differences between effect of mixing methods for 120 minutes, and it is not ture for ultrasonic mixing (USM) according to the Figure 2. It is stated in conclusion again(line 217-218).
6.What is the chemical groups on the surface of rubber particles made by water jet milling? Does any effect on the properties of EP modified by water jet milled worn tire rubber particles caused by those chemical groups on the surface of RP?
7.For Table 1, what is the results of tensile and three point bending tests of rubber filled epoxies containing RP more than 1%?
8.In the abstract, "In this work a cycloaliphatic amine-cured epoxy (EP) resin has been modified by micron scale rubber particles (RP) up to 5 mass %." It should piont out that the preferred usage of RP for EP is 1 mass% for the best of toughening of EP.
Author Response
Please find our response in the attached PDF file.

Reviewer 2 Report
In the present paper, the authors aim at showing that the water jet milling of exhausted tires allows the production of rubber particles which act as toughening agent for an epoxy system.
In the introduction, they assume that this effect is due to the rougher surface of particles produced by water jet milling. To show the rough surface the report in figure 1 micrographs of the milled particles. However, , it is impossible to appreciate the “roughening effect” of the water jet milling, as no other sample obtained from other milling techniques is shown for comparison. – such a micrograph should therefore be added to the paper.
In the materials and methods description, more details about testing should be reported: as an example, at line 128, the authors state that the “Technical average particle size (of water jet milled particles) was about 400 micron”: how was this measured?
Further, indicating the standard number for mechanical testing does not always give enough information about the test and its validity: again, as an example, the ISO 29221 standard for the determination fracture toughness prescribes a series of checks before the results are to be considered valid. They should be discussed – here or in the results section.
Moreover, fracture toughness can be determined following several methods, described by different standards: the authors should motivate the choice to use the crack arrest toughness as a measure of the materials toughness instead of the more commonly adopted toughness at crack onset. Finally on this point, in line 153-154 they state “From the results fracture toughness (KIC) and strain-energy release rate (GIC) was calculated”. However the standard they use give a formula to determine KIC alone, and no indication is given for GIC: how was GIC determined?
Finally, please report standards codes commas (for example standard for fracture toughness measurement should be reported as ISO 29221 instead of ISO 29,221).
In the results subsection on the analysis of influence of mixing parameters on particle size, the authors report the stirring method and stirring time, but do not report nor the stirring speed, nor info on the container and stirrer dimensions (which affect the level of shear strain the material undergoes. Further, the concentration of rubber used in the mixing experiment should be indicated. I also wonder if besides the clear effect of time on particle dimensions, also an effect of mixing speed or dispersion concertation may be significant.
As for figure 3, the authors should clearly indicate which of the two phases (the bright or the dark one) is the particle phase. And at which mixing time refers figure 3c? Further, from figure 3, rubber particles do not seem circular. How was the “particle dimension” defined and measured?
In the result subsection on influence of filler content on mechanical behavior, only the case of mechanical stirring is considered. However, there is no indication of the stirring time (nor that it was the same for all concentrations). Further, no info is given about the particle size and particle size distribution for the three rubber filled epoxy resins considered. This information should be helpful in better justify both the increase of Charpy impact strength with for rubber filled materials and its decrease at increasing rubber contents.
Also the choice of limiting to one filler content (and one stirring time/speed) for the following mechanical characterization is questionable, as it limits the scope of the investigation too much. At least the three amount should be considered for all the characterization. For example, this could help interpreting the results in moduli (both tensile and flexural), confirming the hypothesis stated in line 201 and 202 of the manuscript. As for strengths reduction, do the authors think that also the effect of adhesion at interface may be important?
As for the toughness measurement, the authors observe a significant increase in toughness (measured as KI or a not defined G): however, if the error bars are taken into account, the increase is very limited, if any. In figure 5, the axis related to fracture toughness should be reported as fracture toughness at crack arrest, as indicated in the standard.
In table 1, the unit of measurement of both moduli and strengths should be reported.
In line 190 “unsized rubber” should be corrected in “unseized rubber”?
In line 191 “week” should be corrected in “weak”
In line 199 “Flexural stress” should be corrected in “Flexural strength”. In several points of this section the term stress is misused in place of “ultimate stress” or “strength”, please check and amend.
The conclusions just resume the results and suggest a possible reason for the experimental observations. However, the explanations are mere hypothesis, as no data ere reported to support them. For example, in line 222-223, the statement “Decrease was caused by bigger rubber particles/agglomerates which behave as local stress collecting zones” should be supported by a measure of particles size distribution, and some SEM observation of fracture surface, to locate the point where fracture event initiates. Also the statement “Reason of this behavior is crack deflection may be considered by rubber particles moreover RP retard the crack growing.” (lines 226-228) is reasonable, but has no support from evidences of the paper.
Also the “conclusion” section requires careful read for mistakes/missing words. Just as an example, line 218 “Explanation of this phenomenon is RP create”, probably should be amended in “Explanation of this phenomenon is that RP create”
Author Response

(The authors gave the same response as above.)

Reviewer 3 Report
This paper is well written and in my opinion can be published in current form.
Round 2
Reviewer 1 Report
This revision is reasonable and the significant improvement has been made to give a clear statement that how to toughening of epoxy resin by using water jet milled worn tire rubber particles as additives. It is worthy to be accepted and published in Polymers.
Reviewer 2 Report
Dear authors,
I recognize that the changes and integration to the manuscript made it clearer and more complete.
However, I think the measurement of fracture toughness is questionable, and not acceptable:
1. In the previous version you state the standard adopted for fracture toughness determination was ISO 29221 while in the present version you refer to ASTM D5045. These two standards refer to a completely different setups, and measure the toughness at crack arrest and at crack initiation, respectively. I’m convinced that a full description of the test is required to clarify the test you adopted.
2. Assuming you adopted the D5045, your sample dimension do not comply with the requirements of the standard to ensure that plane strain conditions are met. In particular, given the thickness of your sample, 4mm, the width (and height) of the sample should be at most 4x4=16mm. As you use a width of 50mm you probably are in the transition region between plane stress and plane strain, and you may be in a different condition for the filled and the unfilled system. Given this condition, you cannot compare the results from the two system at all. To address this issue, you should either choose the right geometry (and perform all the tests suggested by the standard) or consider different thickness-to-width ratio and consider check whether KIC depends on the ratio or not. Also in this case you should check that all the geometrical requirements of the standard are met.
3. The formula for GIC (eq 2) does not come from the standard and is wrong – a simple dimensional analysis will made you convinced about it. By the way what is the value of poisson ratio for this system?
Further, using a formula involving KIC and E can be imprecise, as E depends on strain rate, and this parameter may not be the same in tensile test and in fracture test. If you really performed the test according to ASTM standard, you can directly determine GIC from the experimental data.
4. In fracture testing, the way the notch is produced may affect the results. You should give some detail on notching technique and notch length.
As the measurement of material fracture toughness is a focal point of the work, I think that any effort has to be spent to get a meaningful and accurate measurement of this property before publishing the paper.